# Proton PBS Planning Techniques, Robustness Evaluation, and OAR Sparing for the Whole-Brain Part of Craniospinal Axis Irradiation

**DOI:** 10.3390/cancers16050892

**Published:** 2024-02-22

**Authors:** Witold P. Matysiak, Marieke C. Landeweerd, Agata Bannink, Hiska L. van der Weide, Charlotte L. Brouwer, Johannes A. Langendijk, Stefan Both, John H. Maduro

**Affiliations:** 1Department of Radiotherapy, University Medical Center Groningen, 9713 GZ Groningen, The Netherlandsc.l.brouwer@umcg.nl (C.L.B.);; 2Department of Radiotherapy, Mayo Clinic, Rochester, MN 55905, USA

**Keywords:** proton therapy, craniospinal irradiation, treatment planning

## Abstract

**Simple Summary:**

While the dosimetric advantages of proton therapy over conventional treatment for CSI are typically emphasized for spine fields, only lens dose reduction has been reported for brain fields. In this retrospective in silico planning study, the aim is to assess the potential gains of proton CSI for organs at risk in the whole-brain part of CSI. Different optimization techniques and beam arrangements as well as pencil beam spot editing were employed to determine the potential gains. The study shows that: (1) cochlear sparing is impractical, (2) improving the lens dose is possible mainly by adding a posterior field direction, and (3) an incremental scalp dose reduction is possible, particularly in adult patients.

**Abstract:**

Proton therapy is a promising modality for craniospinal irradiation (CSI), offering dosimetric advantages over conventional treatments. While significant attention has been paid to spine fields, for the brain fields, only dose reduction to the lens of the eye has been reported. Hence, the objective of this study is to assess the potential gains and feasibility of adopting different treatment planning techniques for the entire brain within the CSI target. To this end, eight previously treated CSI patients underwent retrospective replanning using various techniques: (1) intensity modulated proton therapy (IMPT) optimization, (2) the modification/addition of field directions, and (3) the pre-optimization removal of superficially placed spots. The target coverage robustness was evaluated and dose comparisons for lenses, cochleae, and scalp were conducted, considering potential biological dose increases. The target coverage robustness was maintained across all plans, with minor reductions when superficial spot removal was utilized. Single- and multifield optimization showed comparable target coverage robustness and organ-at-risk sparing. A significant scalp sparing was achieved in adults but only limited in pediatric cases. Superficial spot removal contributed to scalp V30 Gy reduction at the expense of lower coverage robustness in specific cases. Lens sparing benefits from multiple field directions, while cochlear sparing remains impractical. Based on the results, all investigated plan types are deemed clinically adoptable.

## 1. Introduction

Craniospinal axis irradiation is characterized by high toxicity, predominantly associated with the young age of the patients, the dose to the whole brain, large target volume, and the delivery technique. Some toxicity types, for instance, those related to endocrine and neurocognitive functions, are unavoidable in the CSI phase of the treatment since the pituitary gland and brain are integral parts of the CTV. Consequently, sparing is only possible in the boost phase, although innovative ideas, such as hippocampal sparing CSI, have been investigated [1,2,3]. Due to its physical characteristics, the traditionally used megavoltage photon beam allows for only limited improvements in CSI-related toxicity reduction, even considering the high-technology advances in photon radiotherapy in recent decades [4]. The physical characteristics of protons (i.e., their finite range), on the other hand, allow for the complete exclusion of the oral cavity, dentition, and parotid glands from the radiation field. While CSI implementation for the passive scattering technique mostly focused on target coverage with only limited possibilities of geometrical sparing, PBS optimization techniques enable a better control of the proximal dose, at the same time making it possible to spare organs at risk, such as the scalp, cochleae, and the lenses of the eyes. Due to the proximity of the target, the cochleae are typically included in the PTV; so, a reduction in dose is possible only by jeopardizing the target dose coverage [5,6]. In the robust planning method, however, the target is the CTV, which may allow for an incremental lowering of the cochlea dose while maintaining the clinical target coverage criteria (CTV covered by 95% voxmin IDL), which is possible with a sufficiently steep dose gradient that may be achieved by combining the IMPT type of optimization with more beam directions. The literature reports a threshold scalp dose for radiation-induced alopecia of approximately 21 Gy following proton CSI with high-dose chemotherapy and 30 Gy with conventional chemotherapy [7,8]. Nonetheless, the reduction in the scalp dose is difficult to achieve due to physical properties of proton radiation, i.e., the lack of a skin-sparing effect characteristic for megavoltage beams. Therefore, distributing the entrance dose over more beam directions combined with per-beam dose modulation and the pre-optimization editing of superficial spots may offer a means of incremental reduction in alopecia risk. Compared to the conventional technique, proton CSI can offer dose reduction to the lens of the eye, and the dosimetric benefit may also depend on the choice of beam directions and optimization method [9].

Data on the clinical toxicity outcome comparing protons and photon CSI are scarce. Nonetheless, the recent literature reports on acute toxicity reduction, a better preservation of neurocognitive function, reduced ototoxicity, and a reduced risk of some radiation-induced endocrine abnormalities [10,11,12,13,14,15,16,17]. However, target coverage robustness in proton therapy has been the subject of a longstanding debate, which originates in the observation that, whereas for photons the dose distribution remains fairly invariant to patient anatomy changes and setup errors [18,19,20], for protons, this assumption is no longer valid [21]. As a consequence, the CTV to PTV expansion margins, that for photons provide a buffer zone for these uncertainties, cannot be applied to protons without losing coverage due to the range uncertainty. Therefore, in our clinic, robust treatment plan optimization and evaluation based on the setup and range uncertainties have been clinically adopted.

In this work, we evaluated the potential benefits of (1) the IMPT type of optimization, (2) alternative beam configurations consisting of two and three field directions, as well as (3) the pre-optimization removal of superficial Bragg peaks, against our current clinical practice, which consists of two posterior oblique SFUD-optimized fields and no spot editing. We achieved this by retrospectively generating treatment plans that incorporate a combination of the above techniques, extracting target coverage robustness metrics and OAR doses for RBE = 1.1 as well as for an arbitrary variable RBE model, and comparing them against our current practice. The aim of the study was to determine if any of the techniques are suitable for routine clinical adoption.

## 2. Materials and Methods

This paper is organized in the following way. First, we present our current practice, and then we provide details of the methods and justify the choices we made to arrive at our results by means of comparisons between the different planning options. Section 3 presents the results divided into the discussion of the target coverage robustness and organs-at-risk sparing. These results are further discussed in Section 4.

### 2.1. Cases Included in the Study

Eight consecutive patients, two adults (ages 19 and 65 years) and six pediatric patients (median age of 9 years, ranging from 4 to 14 years), who had previously received craniospinal irradiation at our center and started radiotherapy between the arbitrary dates of 15th May and 4th November 2019, were included in this retrospective study. Five pediatric patients were treated under general anesthesia. The exclusion criteria for the study were the following: (1) the presence of an implanted foreign material in the path of the proton beam (valves, shunts, artificial bone reconstruction materials, and surgical clips; we excluded 3 cases) and (2) an unstable outer contour due to, e.g., seromas or subcutaneous CSF collections (we excluded 5 cases). For the clarity of comparison, all dose statistics presented in this work are normalized to the prescription dose and presented as a fraction thereof.

### 2.2. Definitions of Target and Organs at Risk

CTV-brain, lenses, and cochleas were delineated on a 2 mm CT scan in the bone setting according to the SIOP brain tumor group consensus guidelines [22]. The scalp structure was segmented by creating a 5 mm inner wall from the skin surface covered by hair, and named “Scalp5”.

### 2.3. Optimization and Robustness Evaluation Technique

In our clinical practice, we employ robust optimization and target coverage evaluation features implemented in RayStation ver. 11B (RaySearch Laboratories AB, Stockholm, Sweden). For the whole-brain CTV part of the CSI treatment, we apply a range uncertainty of 3% and an isotropic setup uncertainty of 3 mm.

The robustness of the target coverage against the range uncertainty and translational patient setup errors was evaluated using the method proposed by Korevaar et al. [23]: The voxelwise minimum and voxelwise maximum (voxmax) dose distributions were first created by combining isotropic shifts with proton range uncertainty. Next, to account for the aforementioned perturbations, the voxmin composite dose distribution was reviewed to determine if the target coverage was adequate. Similarly, the voxmax was inspected to ensure that the clinical goals for OARs as well as for the target were met. Our target coverage criterion requires that at least 98% of the CTV volume is covered by the voxmin 94% IDL, while a voxmax hotspot of D2% < 107% is deemed acceptable.

Since the CSI plans consisted of more than a single isocenter, we adopted a position verification protocol in which only 3D table corrections were applied to minimize the misalignment of the patient with respect to the reference scan. As a consequence, our position verification protocol permitted the residual rotational and translational misalignment of maximum values of 2° and 3 mm, respectively, for isocenter 1 (CTV-brain). While tools exist in the TPS to evaluate the robustness of dose distribution against translations, such an evaluation against rotations is not trivial and is not routinely conducted for clinical plans. However, for the purpose of comparing the different plans in this work, an evaluation against rotations was found informative; so, the following methodology was adopted. First, six copies of the planning CT were made with identical structure sets. Next, each copy was rigidly registered to the planning CT with ±2° rotation perturbations along the three cardinal axes and with the pivot point at the 1st treatment isocenter. The arbitrarily chosen ±2° rotations correspond to the tolerance level specified in our position verification protocol. In the following step, a robustness evaluation was performed (i.e., voxmin and voxmax doses were computed) on the planning CT as well as on each of the 6 rotated planning CT sets separately, thus resulting in 7 voxmin and voxmax dose distributions: 1 incorporating the translations and range uncertainty only (the planning set) and 6 additionally including the uncorrelated rotations.

In our clinical practice, isocenter 1 is typically located at the level of the atlas. This determination comes from the available field size considerations; this location allows for covering a long target with a smaller number of isocenters. However, because of the pivot point location, an example 2° yaw or pitch results in a higher misalignment of the CTV than a 2° roll, for which the pivot point is located centrally in the target (see Figure 1). It is then expected that, when 2° rotations are applied, yaw and pitch rotations might result in more dramatic coverage losses than a roll.

### 2.4. Current Practice

In our current technique, we typically use two coplanar, evenly weighted, uniform, lateral oblique fields (typical beam directions are 110° and 250°) to cover the CTV-brain. This technique was adopted on the basis of the following considerations: (1) fields with couch rotations are not preferred due to additional complexity that they introduce to the position verification protocol as well as concerns of reduced robustness of the cervical gradient junction; (2) anteriorly inclined or straight lateral fields do not allow for the geometrical sparing of lenses of the eyes; and (3) posteriorly inclined fields raise concerns of proton end-of-range effects in the optic system. We employed an SFUD type of optimization, which was implemented in the TPS as an optimization parameter requiring that the maximum dose from any field direction must not exceed (107% PD)/(number-of-field-directions). The initial spot placement was determined by the internal TPS algorithm, which considers the desired spot (in-layer) and energy (between layers) spacing as well as the robustness settings. In our practice, we do not edit the initial locations of these spots.

All beams in all investigated plan types used the 4.0 cm WET range shifter, which makes it possible to deliver dose also superficially. Given that the spot size (σ) at the isocenter in air of our delivery system ranged from 6 mm (lowest energy) to 3 mm (highest energy), the use of the range shifter combined with the drift distance necessary to provide safe clearance from the patient and support equipment resulted in approximately double the spot size in air. Finally, for the optimization and final dose calculation, the Monte Carlo algorithm was used.

### 2.5. Candidate Techniques

#### 2.5.1. IMPT vs. SFUD Types of Optimization

While the SFUD optimization is perceived as providing a better target robustness coverage than IMPT, its application limits the use of the physical properties of protons (i.e., the Bragg peak) for the improved sparing of the organs at risk. In this work, both optimization techniques were employed and evaluated against each other in combination with different beam arrangements and the use of superficial spot filtering, with respect to the target robustness coverage and organs-at-risk sparing.

#### 2.5.2. The Number and Directions of Fields

In our current practice, we commonly use the LPO-RPO directions (See Figure 2, left) with the typical gantry directions of 110° and 250°. The posterior oblique directions were selected to provide a balance between the geometrical sparing of the lenses of the eyes while allowing for the adequate coverage of the cribriform plate and avoiding concerns of an end-of-range RBE increase in the brain and optic system. In addition, the wide angle between the fields allows for spreading the entrance dose throughout a larger surface of the scalp, thus helping to reduce the overlap between the entrance doses from both beams. Finally, the use of only two field directions provides a reasonable balance between the plan complexity and the delivery time. The addition of the third (PA) field direction (Figure 2, middle) may be considered with only a small delivery time penalty required for loading and delivering the field, because the required gantry direction is already utilized for the treatment of CTV spine. The 3-field configuration has the potential to further decrease the high scalp dose by reducing the areas where the incident field enters the skin tangentially as well as geometrically helps by sparing the lenses. Furthermore, the robustness of the cervical junction can be improved by modifying the planning technique to transform the cervical junction into a pseudo-junction [24]. The third, non-symmetrical arrangement (Figure 2, right) consists of two field directions only, PA and LPO (or RPO), and can be considered to reduce the delivery time by shortening the time needed to rotate the gantry comparing to the LPO-RPO field arrangement. However, close attention has to be paid to the skin dose, robustness, and the end-of-range effects in the optic system.

#### 2.5.3. Superficial Spot Removal (SR)

The internal algorithm of the robust Bragg peak placement in RayStation does not allow for regions of interest to be strictly used as exclusion zones from spot placement. Instead, such “Avoidance ROIs” also prevent the algorithm from placing the spots downstream of the ROI, effectively serving also as “Blocking ROIs”. In addition, the optimization robustness settings are also taken into account by the algorithm, which results in also placing the spots inside of the avoidance ROIs. To circumvent these issues, an algorithm was created by Hedrick et al. [25] that allows for spot editing after the initial spot placement is conducted by RayStation. In this work, the algorithm was used to remove spots placed less deep than 5 mm from the skin surface (see Figure 3), which follows the clinical practice at the Provision clinic (S. Hedrick, Provision CARES Proton Therapy Center, Knoxville, TN, USA, personal communication). By removing the spots placed superficially, it might be possible to reduce the dose to the skin. However, one also has to ensure that the target can be still adequately covered with an acceptable dose uniformity under uncertainties despite the removal of these spots.

### 2.6. Comparison of the Current Practice with the Proposed Techniques

#### 2.6.1. Ensuring a Fair Comparison between the Different Plan Types

When comparing the different PBS plan types, it is crucial to ensure that the plans are optimized consistently so that the differences can be attributed to the known factors, such as, in this study, the beam arrangement, single- or multifield optimization, and spot editing. For PBS optimization, this is a challenging task, because the optimization problem has a large number of degrees of freedom so that many solutions that satisfy clinical goals are possible and depend on the choice of optimization parameters, their weights, and the path that the optimizer takes to arrive at the solution. The additional difficulty arises when the voxmin composite dose distribution is used as the coverage criterion, because it is only known after the robustness evaluation is completed (i.e., in a separate step following the optimization process). In this work, a method based on the optimization function template was adopted to reduce the variability between the plans that originates in such factors as the selection of the optimization strategy, dosimetrist experience, and the available time for the task. The use of the template helped to highlight the factors that are different between the plans. However, the disadvantage of this approach is that each plan might not be fully optimal and that customization of the optimization parameters could possibly further improve the clinical goals. The template of optimization functions was first created and tested on the available cases so as to ensure that (1) the near maximum dose clinical goals are met, (2) the target coverage is satisfied, and (3) OARs are spared. To achieve the optimal sparing of OARs in different patient geometries, the dose fall-off function implemented in RayStation was used, which effectively acts as a maximum dose objective that linearly changes with the distance between the target and the OAR. To make the distinction between the IMPT and SFUD types of optimization in the optimization template, the uniform dose optimization parameter was applied for the whole plan dose in the IMPT plans, while the per-beam uniform dose optimization parameters with the same weights were requested in SFUD plans.

#### 2.6.2. Target Coverage Robustness Evaluation against Range Uncertainty, Translations, and Rotations

The position verification procedure of craniospinal irradiation adopted at our institution allows for disregarding residual translations as long as the misalignment values are within the tolerance of the position verification protocol. Rotation corrections are not applied, but instead, the patient is re-positioned on the couch—a practice that has limited accuracy.

Although the evaluation of the target coverage robustness against rotations is not routinely conducted in our clinic, it may be used as a tool to compare the robustness of the different plan types. We selected a value of 2° rotations regarding isocenter 1 in each cardinal plane, in addition to the range and patient setup uncertainty (3% and 3 mm, respectively), which are the maximum accepted values in our position verification protocol.

#### 2.6.3. Organs-at-Risk Doses and Variable RBE Evaluations

In this work, we investigated if alternative beam arrangements to our clinical standard, the use of multifield optimization, and the superficial spot filtering provided sufficiently robust target coverage when the uncertainties of the proton range in patient as well as the patient setup (translations and rotations) were taken into account, while offering a better sparing of OARs. To this end, we looked at the CTV voxmin near minimum (D98%) and voxmax near maximum (D2%) DVH parameters [26]. In addition, we designated the maximum point dose parameter (D0.03cc) in the optical structures (chiasm and optic nerves), as a surrogate for dose non-uniformity. The rationale for the choice of these structures is such that the adequate coverage of the neighboring cribriform plate is typically challenging due to the presence of tissue inhomogeneities (air cavities and bony structures of the paranasal sinuses) and competes with sparing the lenses of the eyes [27]. For this reason, hotspots may easily arise in this part of the target when setup and range uncertainties are considered, especially when the dose per beam is not uniform. In order to improve the sensitivity of the analysis, we recomputed the plan dose using a variable RBE model that takes into account the LET dependence of the deposited dose on RBE. We arbitrarily chose to use the McNamara variable RBE model [28] with α/β = 2 Gy for this task.

## 3. Results

### 3.1. Robustness of the Target Coverage

Figure 4a shows the near minimum (D98%) voxmin dose values for CTV-brain with only range and translational setup uncertainties accounted for. In all cases, the voxmin 94% isodose line covers at least 98% of the CTV-brain volume, which is our clinical target coverage acceptance criterion. Note that, for all plans where the superficial spot removal was applied, the common template that was used to optimize all cases resulted in a consistently lower target coverage by a median of 0.6% PD. When ±2° rotations about the three cardinal axes were considered in addition to translations and range uncertainty (Figure 4b), the coverage was further impacted and, in a few instances, reduced to below the acceptance level. Notably, for cases 3 and 5, the median coverage for the two plan types SFUD-(LPO-PA)-[SR] and IMPT-(LPO-PA)-[SR] approached the threshold. In both of these cases, the CTV was located in a shallow position below the skin because of thin cranial bones and, as a result, the 5 mm structure used to remove spots was partially overlapped with CTV-brain, thereby removing the spots located on its peripheries. Both plans used the superficial spot removal as well as two field directions, which included one straight posterior field. When examining the individual dose distributions, we noticed that the most significant loss of coverage occurs under ±2° yaw rotations and was caused by the use of the posterior field (see Figure 1, which shows the effects of rotations). We conclude that the use of superficial spot removal with a low separation distance between the scalp and CTV-brain in combination with the LPO-PA field arrangement are the main factors contributing to the reduced coverage.

The CTV-brain D2% metric was designated as a measure of the dose homogeneity. As presented in Figure 5, all plans met the near maximum dose criterion of voxmax 107% PD. The results show only small variations in D2% between the plans as well as a consistent difference between voxmax and the nominal doses of approximately 1.7% PD. When accounting for 2° rotations in addition to translations and range uncertainty, the D2% metric showed only small variations; see Figure A2 and Figure A3.

In addition to CTV-brain D2%, the maximum dose to the optic structures was designated as a surrogate for coverage uniformity under setup and range uncertainties in the region where extreme density inhomogeneities are present (cortical bone and sinus cavities) as well as where competing objectives are requested: the adequate coverage of the cribriform plate and the sparing of the lenses. The same methodology was applied for the analysis of the near maximum dose, but in addition to the RBE = 1.1 model, the variable RBE model of McNamara was used. Figure 6 presents the McNamara D0.03cc dose to the optic structures for all cases and plan types but excluding the setup uncertainties. The D0.03cc evaluated on RBE = 1.1 dose distribution as well as the effect of rotations are presented in the Appendix A in graphical and tabular forms (Figure A4). Based on this analysis, we conclude that, while the point doses for the different plans vary from the reference plan, the variations are small even when considering the variable RBE due to the proton end-of-range. In addition, the maximum dose to the optic structures does not considerably change when the rotations are accounted for.

### 3.2. Organs-at-Risk Sparing

#### 3.2.1. Lens of the Eye

All alternative beam arrangements showed statistically significant mean dose reductions to the lens of the eyes by 2% to 7% PD with respect to the reference plan type (Figure 7). For most plan types and cases, this difference was also carried over when considering voxmax mean doses, except for a few notable exceptions, namely, the dose to the right lens in the LPO-PA field arrangement increased considerably between the nominal and voxmax doses. This can be explained by the high dose gradient that is created by the distal fall-off of the LPO field to spare the lens in the nominal plan.

#### 3.2.2. Cochleae

As expected, only small differences between cochlear doses for the different plans and cases were present, and the differences between the voxmax value and nominal doses were consistent (2% to 3% PD) (Figure 8). Furthermore, the size of the spot, presence of inhomogeneities (bone, soft tissue, and air), and the proximity of the target to the CTV did not allow for creating a dose gradient that allows sparing. As a result, regardless of the plan type, the differences in cochlea doses were very small. The lack of a significant dose gradient means also that sparing the cochlea does not depend on the optimization type (e.g., PTV-based or CTV-based robustness).

#### 3.2.3. Scalp

The optimization of the scalp dose made use of the generalized equivalent uniform dose [29] with the value of the exponent of 10 as presented in the publication of Min et al. [7] to preferentially reduce the high dose component. In accordance with the publication of Min et al. [7], V21Gy and V30Gy were extracted and evaluated for the different plans and cases. This constitutes a deviation from presenting the data in this work as fractions of PD; so, for this presentation, the PD was assumed to be 36 Gy.

Based on Figure 9, it is apparent that V30 Gy is much lower for cases 1 and 2 (adults) than for the remaining cases (pediatric patients). Although less pronounced, this difference is also present for V21 Gy (see Figure A11: V21 Gy to Scalp5). The reason for this difference is the presence of the larger separation between the CTV-brain and skin surface, which in adults, is provided by thicker cranial bones and subcutaneous tissue, sufficient for the distal fall-off dose gradient, while for pediatric patients, the separation is too small to allow sparing while providing adequate CTV coverage (Figure 10). 

In the two adult cases, the larger distance between the CTV and skin also makes the scalp sparing using superficial spot removal slightly more effective than in the pediatric cases. When evaluating the DVHs of the scalp structure individually for all cases (Figure A12), it is apparent that the superficial spot removal is the most effective mechanism of scalp dose reduction (optimization method and beam arrangement are of lesser importance). The other aspect is the difference between the nominal and voxmax doses: Figure A10 and Figure A11 show V30Gy and V21Gy, respectively, for Scalp5, which illustrates that the difference between the two is higher for the SR plans than for the non-SR plan. However, the dose reduction is still favorable for the SR plans.

From Figure 9, it is also clear that superficial spot removal is effective in reducing V30 Gy to the scalp in both nominal as well as voxmax by a median of 9% PD and 3% PD, respectively, although the difference between the voxmax and nominal doses is higher for the plans that utilize superficial spot removal.

The reduction shown in the figures as well as the dose volume histograms can be also illustrated by superimposing the isodoses onto the skin surface contour (Figure 11). The differences in the level of scalp sparing between the adult and pediatric cases are apparent. Furthermore, the effect of the superficial spot removal is visible in both cases. Finally, the redistribution of the high dose area toward the posterior section is visible when the third (posterior) field is added.

The shape of the high dose region in the adult case (notice the 80% IDL in Figure 11, two-field beam arrangements) follows the shape of the entrance dose, while for the pediatric case (see the 90% IDL), it is more consistent with the exit dose. This observation illustrates the contributions of entrance and exit doses to the total scalp dose. The addition of the third field direction also redistributes the high dose region more uniformly to the dorsal part of the scalp.

## 4. Discussion

The current study comprises of two parts: in the first one, we investigated the CTV-brain coverage robustness resulting from all evaluated plan types, while the second part constitutes an attempt to quantify the dosimetric sparing of lenses, cochleae, and the scalp.

As demonstrated in Figure 4a, all investigated plan types met the internally adopted voxmin D98% > 94% target coverage criterion. After factoring in the ±2° rotations about the cardinal axes (Figure 4b), the target coverage was still acceptable for most plan types; however, for some cases, the coverage deteriorated. Notably, for the LPO-PA beam arrangement, it was reduced more significantly than for the other plan types. which is mostly a consequence of a combination of two factors: the use of superficial spot removal and the low separation between the CTV and the skin. In addition, the target coverage from the posterior field is less robust against yaw rotations when the isocenter is placed not centrally inside the CTV-brain. This shortcoming, however, can likely be helped by shifting the isocenter location to a more cranial position. It is also apparent that the use of superficial spot removal results in a consistently lower coverage than for the corresponding plans that do not use superficial spot removal. In addition to coverage degradation, the use of SR results in higher near maximum voxmax values for CTV-brain (Figure 5), which indicates that some spots removed by the script were necessary to achieve a more uniform target coverage. The balance between the near minimum and near maximum doses is a result of the arbitrary choice of functions that constitute the optimization function template, which was applied to all cases and plan types to facilitate the comparison consistency of the different plans. It is likely to expect an improved target coverage if a higher near maximum dose is accepted. The near maximum CTV-brain dose was only weakly varying with rotations in addition to translations and range uncertainty (Figure A2 and Figure A3) for each plan type. Therefore, in order to increase the sensitivity of the analysis, we also looked at the dose to the optic structures, which might potentially be affected by proton end-of-range, and used a smaller volume (0.03 cc) in combination with a biological dose model in which the RBE depends on the LET distribution as well. The combination of these factors helped us to highlight the differences between the different plan types. We chose the optic structures because an increase in the biological dose might have clinical relevance. Figure 6 illustrates the result of this investigation and Figure A4 and Figure A5 provides a quantitative analysis: while some plan types result in D0.03cc dose to the optic structures to be statistically higher than the reference plan, the differences are small (the median increase is limited to approx. 3% of PD in the variable RBE evaluation).

It is apparent from the literature that, among the OARs located in the head, proton CSI offers a considerable dose reduction mainly to the lens of the eye. The clinical significance of the lens dose was reported in a study of childhood cancer survivors [30], in which a linear dose–response for an increased risk of cataracts was found in individuals who were exposed to 0.5 Gy or more to the lens of the eye. The odds ratio for developing cataracts was found in that study to increase with the average dose to the lens at a rate of 1.53% per Gy. Given that the typical dose to the lens of the eye is estimated at 15–35% of the PD [9], the excess risk for developing cataracts following CSI treatment is considered high. In this work, we demonstrated that the alternative beam arrangements evaluated in this study resulted in lower doses to the lens by approximately 2% to 7% of the PD with respect to the LPO-RPO arrangement. The most benefit to the lens dose is visible in plans that utilize three field directions. The asymmetrical LPO-PA beam arrangement results in a left lens dose that is consistent with that of the three-field plans, while the right lens dose is reduced only moderately, showing that sparing the lens using the distal fall-off that is degraded by passing through the inhomogeneous region of the paranasal cavities is less effective than that provided by the lateral penumbra. This asymmetrical beam arrangement could be partially symmetrized by alternating field directions every other day, i.e., LPO-PA and RPO-PA. It is also worth noticing that the median difference between the voxmax value and the nominal doses to the right lens dose (Figure A7) is higher for plan types using two rather than three field directions.

We posed a hypothesis that, by making use of the IMPT type of optimization, a sufficiently steep dose gradient could be created between CTV and cochleae, which would allow for an incremental reduction in the cochlear dose while still providing a robust coverage of the CTV. However, the data analysis revealed no difference in the dose to cochleae between the SFUD and IMPT type plans. The larger spot size, the presence of density inhomogeneities, as well as the cochlea delineation directly adjacent to the CTV-brain did not allow for creating a cochlea-sparing dose gradient.

Two main factors contribute to the reductions of V30 Gy and V21 Gy to the scalp: a sufficient separation between the CTV and the skin as well as the use of superficial spot removal. The former makes it possible to achieve a V30 Gy to the scalp of approx. 1/3 of the volume in adults, while the latter makes it possible to reduce V30 Gy further by an additional 10% in adults and 8% in pediatric patients (Figure 9 and Figure A10). Although the quoted reductions for adult patients are not statistically significant because only two cases were included in this study, they are consistent for all plan types.

Ensuring a fair comparison between PBS plans is a difficult task and the literature does not provide a consistent methodology. The task is even more complex when the target coverage is not a parameter that results directly from optimization but only follows the next step, which is the robustness evaluation, where the voxmin composite dose distribution is computed. In this work, we set out to ensure the consistency of the optimization process by creating a universal template of optimization functions that was applied to all cases. The limitations of this approach are that (1) the list of functions comprising the template is arbitrary and (2) the final plans might not be fully optimal. So, while the consistency of the optimization process highlights the differences between the plan types, the values of the sparing potential for OARs should not be interpreted in the absolute sense but rather as relative to the other plan types.

## 5. Conclusions

This study demonstrated that all investigated beam arrangements provide a similar robustness of target coverage, although in patients with a low separation between the CTV and the scalp, two factors reduce the coverage robustness: the use of LPO-PA field arrangement as well as superficial spot removal. We also observed that the variable RBE shows only a small and typically insignificant increase in D0.03 dose to the optic structures when IMPT is used, the posterior field is added, or both. Sparing the lenses is effective with the straight posterior field, while an insufficient gradient was created to even incrementally spare the cochleae. Finally, a substantial scalp sparing (V30 Gy) is possible in cases with sufficient separation between the CTV-brain and the scalp (i.e., adults), while a reduction with superficial spot removal is effective in all patients.

## Figures and Tables

**Figure 1 cancers-16-00892-f001:**
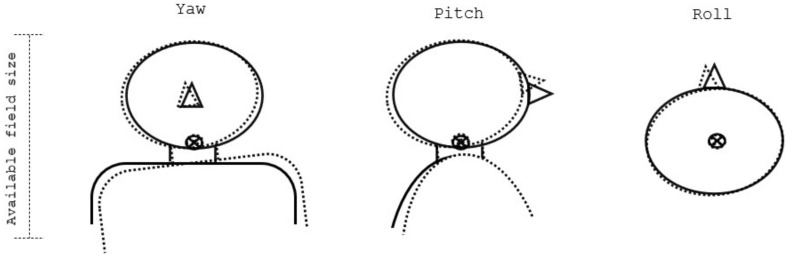
Visualization of the rotations along the cardinal axes when the isocenter (depicted by the ⦻ sign) is placed at the level of the atlas.

**Figure 2 cancers-16-00892-f002:**
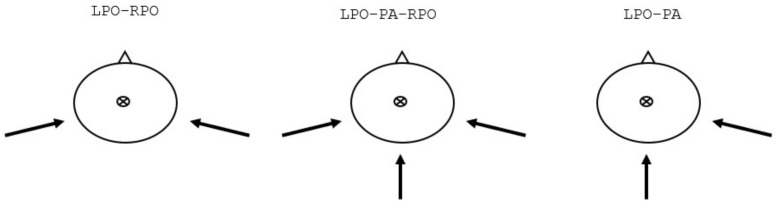
Field arrangements evaluated in this study.

**Figure 3 cancers-16-00892-f003:**
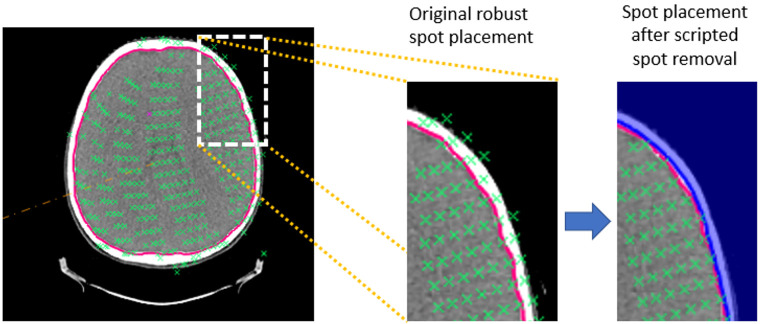
Illustration of the superficial spot removal action. Locations of Bragg peaks (green crosses) originally placed by the TPS to cover CTV-brain (red) by the RPO field. By the action of the superficial spot removal, the Bragg peaks located less deep than 5 mm from the skin surface (blue) were deleted.

**Figure 4 cancers-16-00892-f004:**
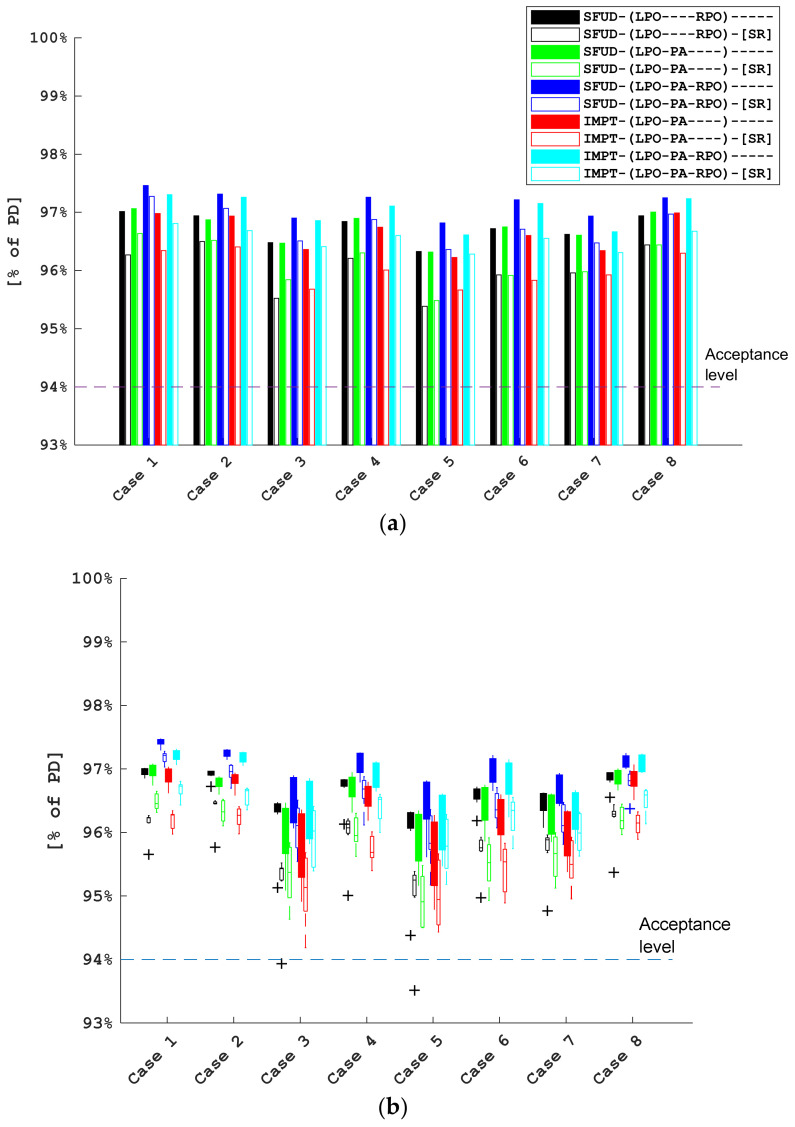
D98% voxmin CTV-brain coverage for the 8 cases and 10 plan types. The legend provides the color code for the different plan types in the format: Optimization Type-(FieldArrangement)-[SpotRemoval]. Panel (**a**): only 3% range and 3 mm translational setup uncertainties included. Panel (**b**): 2° rotations independently evaluated about each cardinal axis in addition to range and translational uncertainties. Boxes: the bottom and top edges of the box indicate 25th and 75th percentiles, respectively; the whiskers show the minimum and maximum values that are not considered outliers; and the ‘+‘ signs show the outliers (i.e., data points located more than 1.5 × interquartile range below the first or above the third quartile).

**Figure 5 cancers-16-00892-f005:**
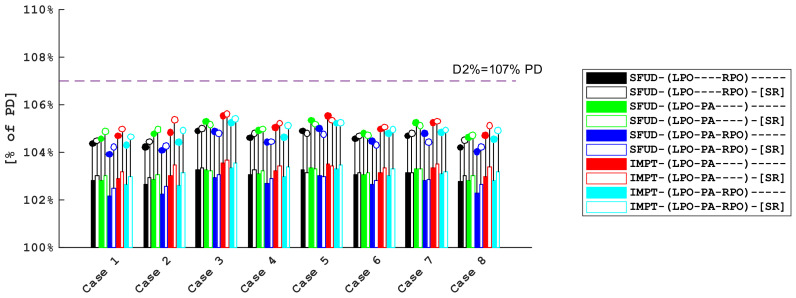
CTV-brain D2% per case and plan type: bars represent nominal doses and dots represent voxmax values.

**Figure 6 cancers-16-00892-f006:**
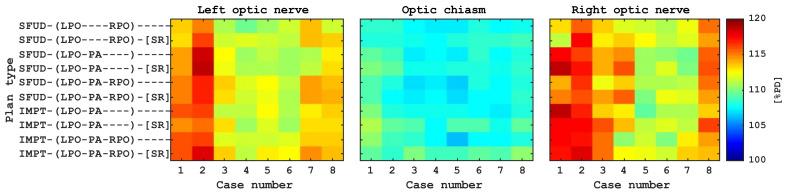
McNamara D0.03cc dose to the optic structures for the different cases and plan types. See the Appendix A for numerical data.

**Figure 7 cancers-16-00892-f007:**
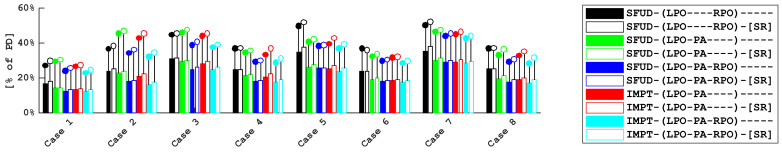
Mean dose to the right lens for the different cases and plan types. See Figure A6 and Figure A7 for the graph showing the left lens and numerical data for both lenses.

**Figure 8 cancers-16-00892-f008:**
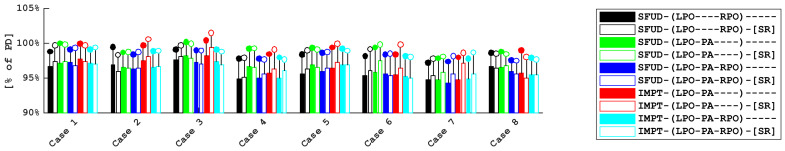
Mean dose to the right cochlea for the different cases and plan types. See Figure A8 and Figure A9 for the graph showing the left cochlea and all numerical data for both cochleae.

**Figure 9 cancers-16-00892-f009:**
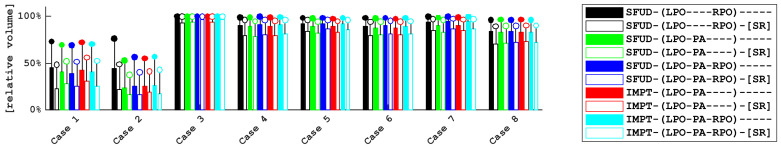
V30 Gy dose to scalp, assuming a prescription dose of 36 Gy. Bars show the nominal dose and circles indicate the voxmax value. See Figure A10 and Figure A11 for V21 Gy and data in tabular form.

**Figure 10 cancers-16-00892-f010:**
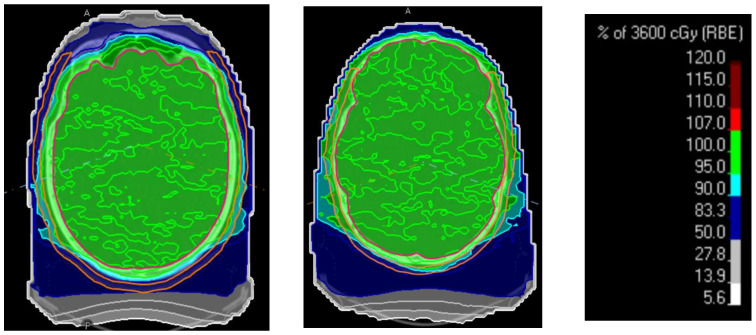
Illustration of the separation between the CTV-brain (red) and scalp (orange): adult case (left) and pediatric case (right).

**Figure 11 cancers-16-00892-f011:**
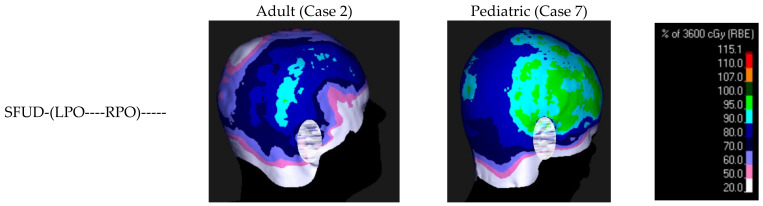
Isodoses superimposed on the body contour for the adult case (left) and the pediatric case (right) to qualitatively illustrate the differences in dose distribution between the 3 selected plan types (ear lobe blurred to prevent patient identification).

## Data Availability

The numerical data presented in this study are available upon request from the corresponding author. Dose distributions in the volumetric images are not available due to patient privacy concerns.

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
