# Peer review of "Proton PBS Planning Techniques, Robustness Evaluation, and OAR Sparing for the Whole-Brain Part of Craniospinal Axis Irradiation"

_cancers, 2024, doi:10.3390/cancers16050892_

Round 1

Reviewer 1 Report

Comments and Suggestions for Authors

This is a very carefully conducted dosimetry study. It brings an interesting approach to the patient setting. It is very important to address this evaluation, but I would suggest its publication in a lower impact journal. On the other hand, it might help the others to bridge the problems of planning and patient setup in such a complicated situation as craniospinal axis irradiation.

Author Response

We appreciate the honest comment provided by the reviewer. This dosimetric study reviews different approaches that various institutions adopt to treat CSI and compares them in a quantitative manner. For the institutions that currently use protons it might provide a material to evaluate their approach as well as help new users to decide on implementation of the technique. Given the utility of proton therapy for CSI, we advocate to give this work adequate exposure.

Reviewer 2 Report

Comments and Suggestions for Authors

The paper discusses proton PBS planning techniques, specifically for Craniospinal Irradiation (CSI). Proton therapy is an effective treatment for CSI, as it provides better dosimetric outcomes compared to traditional treatments. However, while there has been a lot of attention given to spine fields, the lens of the eye is the only area that has seen a reduction in dose for brain fields. This study aims to evaluate different treatment planning techniques for the entire brain within the CSI target, to determine if it is feasible and if there are any potential gains to be made.

According to the study, all beam arrangements provided similar coverage for the target, except for the LPO-PA field arrangement and superficial spot removal in patients who have less separation between CTV and scalp. The use of IMPT had a minimal effect on optic structures with variable RBE. The straight posterior field was effective in protecting the lenses but not the cochleas. Scalp sparing was possible in cases where CTV-brain and scalp had enough separation.

The paper is well-organized. The introduction covers relevant issues, methods are clearly written, results are detailed in tables and graphs, and conclusions are supported by findings.

Author Response

We would like to thank the referee for their comments.

Reviewer 3 Report

Comments and Suggestions for Authors

Title:

The authors provided a paper about "Proton PBS Planning Techniques, Robustness Evaluation, and OAR Sparing for Whole Brain Part of Craniospinal Axis Irradiation".

Abstract:

The authors state "Lens sparing benefits from multiple field directions, while cochlear sparing remains impractical" but later on they state that "Based on the results, all investigated plan types are deemed clinically adoptable".

How could all plans be put at the same clincical level if the cochlear sparing turns out to be impractical? Please clarify

Text:

I would suggest that the author do not use abbreviations in the title such as PBS and OAR since the JOurnal covers all aspect of oncology and not only radiotherapy therefore some readers may not understand such abbreviations.

I would recommend that the authors do not just compare the proposed techniques with their "current practice" but it would be important to state wheter the "current practice" relies on previous literature data (with accompaning references) or it is just "current practice" (in this case please discuss more in detail the clinical choice).

It would be important to justify the choice to investigate the three organs at risk proposed namely lenses, cochleas, and scalp (why did the authors choose to exclude optic nerves for example?).

Round 2

Reviewer 3 Report

Comments and Suggestions for Authors

The authors have satisfactorily addressed my previous comments